# Towards Faithful Sign Language Translation

## Abstract

Sign language translation (SLT) aims to translate perceived visual signals into spoken language. Recent works have achieved impressive performance by improving visual representations and adopting advanced machine translation techniques, but the faithfulness (*i.e.*, whether the SLT model captures correct visual signals) in SLT has not received enough attention. In this paper, we explore the association among SLT-relevant tasks and find that the imprecise glosses and limited corpora may hinder faithfulness in SLT. To improve faithfulness in SLT, we first integrate SLT subtasks into a single framework named MonoSLT, which can share the acquired knowledge among SLT subtasks based on their monotonically aligned nature. We further propose two kinds of constraints: the alignment constraint aligns the visual and linguistic embeddings through a sharing translation module and synthetic code-switching corpora; the consistency constraint integrates the advantages of subtasks by regularizing the prediction consistency. Experimental results show that the proposed MonoSLT is competitive against previous SLT methods by increasing the utilization of visual signals, especially when glosses are imprecise.

## 1   Introduction

Sign languages, as a typical visual language, fulfill the same social and mental functions within the Deaf community effectively. Sign languages convey information through a unique physical transmission system and the corresponding linguistic theory [1], which makes them differ greatly from spoken languages. To bridge the communication gap between the Deaf and hearing communities, vision-based Sign Language Recognition (SLR) [2, 3] and Sign Language Translation (SLT) [4–6] have attracted much attention over several decades. Recent works often evaluate different aspects of sign language understanding models on these two tasks: the effectiveness of the feature extraction [7–9] and the transferability from visual features to the target spoken language [10–12]. However, the association between these two tasks has not been paid enough attention.

Gloss[1] sequences play a critical role in both SLR and SLT. On one hand, recent SLR datasets [3, 11] have limited samples and only provide sentence-wise annotations (i.e., gloss sequences) due to the high cost of frame-wise annotations, and the monotonous alignment between the gloss sequence and sign clips makes it possible to leverage Connectionist Temporal Classification (CTC) [13] to provide supervision. On the other hand, Gloss sequences are widely used as the input of Gloss2Text (G2T) task to estimate the upper bound of Sign2Text (S2T) task [6, 12]. The relationship among these tasks is illustrated in Fig. 1(a). Because glosses can be used to evaluate both SLR and SLT models, it is logical to assume that the SLR model with a lower error rate (more accurate prediction of glosses) can provide more accurate translation results. However, as a visual language, sign language conveys information through multiple visual signals and glosses are imprecise representations of sign videos [10]. Many attempts [10, 14–16] have been done to improve the visual representations but how to reduce effects from imprecise gloss has not attracted enough attention.

---

[1]Gloss is the written approximation of a sign.

Submitted to 37th Conference on Neural Information Processing Systems (NeurIPS 2023). Do not distribute.

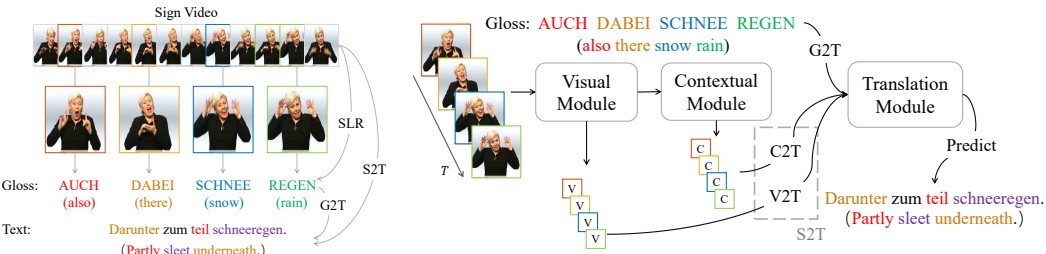

(a) Illustration of SLR, S2T and G2T      (b) llustration of different subtasks of SLT

Figure 1: (a) An example from Phoenix14T [3]. The goal of SLR is to recognize a gloss sequence, which is monotonically aligned with sign clips, from the sign video. S2T and G2T aim to translate sign videos and gloss sequences into spoken language sentences, and G2T is often regarded as the 'upper bound' of S2T. (b) We decompose S2T into two subtasks based on the temporal receptive fields of source features: Vision-to-Text (V2T) and Context-to-Text (C2T), all SLT subtasks have monotonically aligned source features.

As shown in Fig. 1(a), S2T and G2T are similar translation tasks. Different from general multilingual Neural Machine Translation (NMT) [17, 18], they have monotonically aligned source languages (glosses and sign clips) and the same target language. Previous works attempt to improve SLT performance by adopting large-scale pretrained LMs [12] and leveraging extra corpus [11, 19]. These works are developed under the paradigm that 'improves G2T first and then transfers to S2T', which greatly improve S2T performance but inevitably face the hallucination problem [20] (*i.e.*, S2T models tend to generate fluent but inadequate translations), and we attribute this problem to the lack of faithfulness [21] (*i.e.*, the S2T models cannot capture correct visual signals). Besides, the availability of G2T corpora is also the bottleneck for the generalization of the pretrained model.

In this paper, we attempt to increase the utilization of visual signals in S2T to improve faithfulness, especially when glosses are imprecise. We first decompose S2T into two subtasks based on the temporal receptive fields of source features: Vision-to-Text (V2T) and Context-to-Text (C2T). As shown in Fig. 1(b), from V2T to C2T to G2T, the degree of visual abstraction of source features gradually increases, while the translation quality will get better generally. We revisit recent SLT approaches [22, 12] and observe that it is hard for V2T models to find the corresponding visual clips during training, while this is exactly the strength of G2T models. Moreover, improving the alignment between visual clips and target words can improve the faithfulness of translation and relieve the hallucination problem. Different from recent works [11, 12, 19] that attempt to improve the 'upper bound' (G2T) of C2T, we focus on the association among these tasks and try to improve the 'lower bound' (V2T) of C2T.

Specifically, we first integrate the learning of SLT subtasks into a single framework named MonoSLT by sharing their translated modules, which can share the acquired knowledge among SLT subtasks based on their monotonically aligned nature. We further propose two kinds of constraints to enhance faithfulness in SLT. The alignment constraint implicitly aligns the visual and linguistic embeddings through the shared translation module and synthetic code-switching corpora, which are generated by replacing partial visual embeddings with their corresponding gloss embeddings. The consistency constraint regularizes prediction consistency between different subtasks, which can improve both training efficiency and translation quality. Experimental results show that the proposed approach can surpass previous SLT methods on Phoenix14T by increasing the utilization of visual signals.

Our contributions can be summarized as follows:

◇ Exploring the association among different relevant tasks about SLT and integrating SLT subtasks into a single framework named MonoSLT, which can share the acquired knowledge among SLT subtasks based on their monotonically aligned nature.

◇ Proposing two kinds of constraints to enhance faithfulness in SLT. The alignment constraint aligns the visual and linguistic embeddings through a shared translation module and synthetic code-switching corpora, and the consistency constraint leverages the advantages of subtasks by regularizing the prediction consistency.

◇ Showing the lack of faithfulness in recent SLT methods and verifying the effectiveness of MonoSLT for the utilization of visual signals, especially when glosses are imprecise.

## 2 Related Work

**Sign Language Translation.** With the development of vision and language understanding algorithms, SLT has progressed rapidly in recent years [6, 23, 22, 11, 12, 15]. Recent SLT methods can be roughly categorized into two categories: vision-based and language-based.

Vision-based SLT works devote to learning useful visual representations from videos. Considering the relationship with SLR, recent SLT solutions can be roughly divided into three categories: SLR-pretrained, SLR-supervised, and SLR-free. SLR-pretrained solutions initialize the visual extractor with pretrained SLR models [12, 15] or directly adopt the pretrained SLR models to extract visual embeddings [10, 19]. SLR-supervised solutions [22, 14, 15] adopt the multi-task framework and leverage the supervision from both SLR and SLT. SLR-free solutions [23–27] attempt to tokenize visual information without gloss supervision and leverage more real-life data. Recent empirical results [15, 19] indicate that adopting more accurate SLR models in SLR-pretrained and SLR-supervised solutions often leads to better translation quality, but little work has been done to investigate the association between them.

On the other side, language-based SLT works focus on the linguistic difference between sign languages and spoken languages. The pioneering work [6] regards SLT as a typical NMT task and shows the potential of the encoder-decoder framework. Joint-SLRT [10] further adopts the transformer architecture [28] to integrate both SLR and SLT into a single framework. However, it is costly to collect large amounts of parallel corpora for SLT and recent works [11, 12, 19] reveals that data scarcity hinders the further development of SLT. To relieve this problem, Zhou *et al.* [11] leverage rich monolingual data and adopt back-translation to generate synthetic parallel data as a supplementary. Chen *et al.* [12] explore the potential of denoising auto-encoder that pretrained on large-scale multilingual corpora and progressively pretrain each task to achieve effective transfer in SLT. SLTUNet [19] proposes a unified model for multiple SLT-related tasks to further improve the translation. Our motivation is similar with [19] but we focus more on faithfulness, and leverage the monotonically aligned nature of SLT subtasks to align visual and linguistic embeddings.

**Faithfulness in NMT.** With the rapid development of the NLP techniques [28–31], the robustness and interpretability of NMT systems become a crucial issue. A good NMT model should produce translations that capture the intended meaning of the source language (faithfulness) while maintaining grammatical correctness and naturalness in the target language (fluency) [32, 33]. However, NMT models may generate hallucinations due to exposure bias [34], domain shift [35], lack of coverage [36], and other factors [20]. To enhance the faithfulness in NMT, Tu *et al.* [36] maintain a coverage vector to encourage NMT models to consider more source words, Wang and Sennrich [35] leverage minimum risk training to mitigate domain shift, and Feng [33] propose a faithfulness part to enhance the contextual representation of encoder output. Different from general NMT tasks, SLT models need to encoder source information from unsegmented video, which makes it harder to learn the correspondences between video and language and generate faithful translations. We focus on the relationship between visual and language translation tasks rather than diving into specific translation module designs, which makes the proposed method compatible with other NMT techniques.

## 3 Approach

In this section, we first introduce notation and background knowledge briefly. Then we explore the association among SLT-relevant tasks and present some empirical findings about the lack of faithfulness in SLT. After that, we propose a method to improve faithfulness in SLT.

### 3.1 Background

Formally, given a sign sequence $\mathcal{X} = \{\boldsymbol{x}_1, \cdots, \boldsymbol{x}_T\}$ with $T$ frames, SLR aims to recognize its corresponding gloss sequence $\mathcal{G} = \{g_1, \cdots, g_N\}$ with $N$ glosses ($N \leq T$ in general), which are monotonically aligned with sign clips $\mathcal{S} = \{\boldsymbol{x}_{\eta_1}, \cdots, \boldsymbol{x}_{\eta_N}\}$ and $\eta_i$ is the corresponding frame indexes of gloss $i$. The SLR model is generally optimized by CTC, which leverages all possible alignments between $\mathcal{X}$ and $\mathcal{G}$ and can be written as $L_{CTC} = -\log p(\mathcal{G}|\mathcal{X})$. Different from SLR, the objective of SLT is to translate $\mathcal{X}$ into spoken language sentence $\mathcal{W} = \{w_1, \cdots, w_M\}$ with $M$ words ($M \neq N$ in general), which often has different grammar and vocabulary, and the SLT model is optimized by minimizing the negative log-likelihood $L_{SLT} = \sum_{t=1}^{M} -\log p(w_t|\mathcal{X}, w_{<t})$.

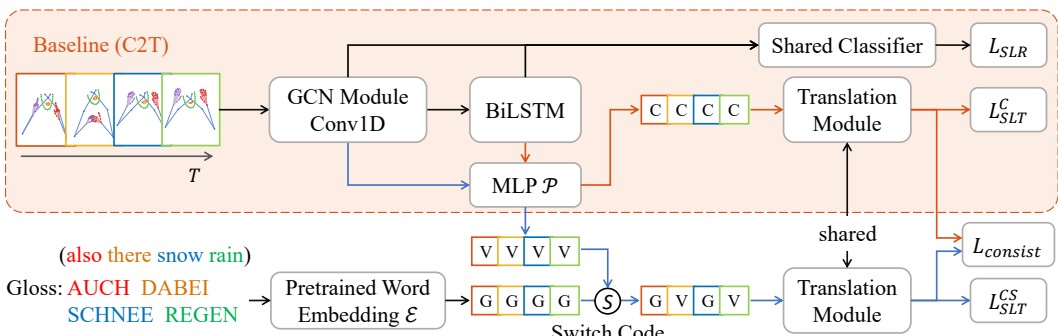

Figure 2: Overview of the proposed method. For baseline (C2T), the visual module is composed of a lightweight GCN-based module and a Conv1D module, and the contextual module is implemented as a two-layer BiLSTM. The proposed method has an auxiliary branch that takes the switched gloss and visual embeddings as input, and both branches share the same translation modules. An additional consistency loss is adopted to regularize the prediction consistency.

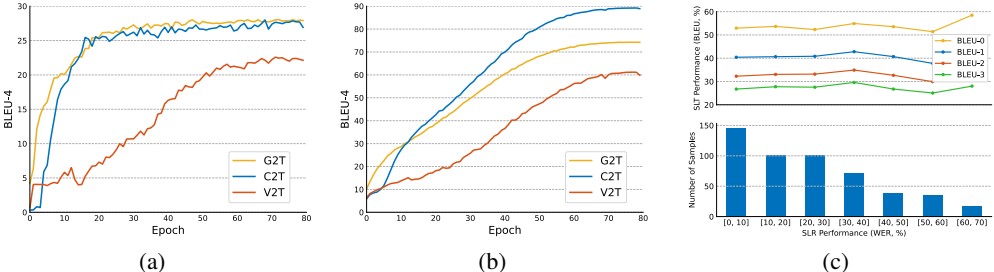

|            (a)            |            (b)            |            (c)            |

Figure 3: BLEU-4 scores of different subtasks over epoch on Phoenix14T (a) dev and (b) training sets. (c) Fluctuation of SLT performance over SLR performance (the upper), and the corresponding number of samples for each SLR performance interval (the lower) on Phoenix14T dev set.

Recent SLT architectures [14, 9, 12] commonly contain three components: a visual module, a contextual module, and a translation module. The basic architecture used in this paper is visualized in Fig. 2. Considering the training efficiency, we use the coordinates of keypoint sequences as inputs. As for the visual module, we adopt a lightweight GCN-based module and a two-layer temporal convolution block (Conv1D). The outputs of the visual module $\mathcal{V} = \{\boldsymbol{v}_1, \cdots, \boldsymbol{v}_T\}$ are fed into a two-layer BiLSTM to obtain contextual features $\mathcal{C} = \{\boldsymbol{c}_1, \cdots, \boldsymbol{c}_T\}$. As mentioned in Fig. 1(b), all of $\mathcal{V}$, $\mathcal{C}$, and $\mathcal{G}$ can be used as the source language for SLT, which are corresponding to V2T, C2T, and G2T subtasks, respectively. Similar to VAC [9], we adopt two classifiers on both $\mathcal{V}$ and $\mathcal{C}$ to provide supervision for SLR, and the basic supervision can be formulated as:

$$L_{basic} = L_{CTC}^{\mathcal{V}} + L_{CTC}^{\mathcal{C}} + \lambda_{\mathcal{C}} L_{SLT}^{\mathcal{C}}, \tag{1}$$

where the superscript indicates the input features of the loss function and $\lambda_{\mathcal{C}}$ is the translation weight.

### 3.2 Exploring the Association among SLT-relevant Tasks

As shown in Fig. 2, the adopted baseline can learn SLR and SLT jointly, and the features of SLR are further utilized by the translation module, which provides a sufficient basis to explore the relationship between different SLT-relevant subtasks. We first train three individual models for V2T, C2T, and G2T, respectively, and visualize the evaluation results during the training on Phoenix14T [3] in Fig. 3(a) and 3(b). We can observe different convergence behaviors on SLT subtasks: the G2T model converges faster at the beginning and achieves higher performance on the dev set, the S2T model achieves comparable performance on the dev set but tends to overfit the training set, while the V2T model encounters difficulties in converging. This observation indicates that the C2T model meets the issue of overfitting before finding the correct visual signals, especially when adopting a powerful translation module, and we identify this issue as the lack of faithfulness.

Moreover, we divide the dev set into several subsets based on SLR performance, and visualize the relationship between SLT and SLR performance of the C2T model in Fig. 3(c). It is surprising

to observe that there is no significant negative correlation (*i.e.*, achieving higher BLEU scores on the subset with lower WER) between the performance of SLR and SLT, even though lower WER indicates the less accumulated error. We analyze results and find that C2T models tend to generate hallucinations [20], which are fluent but unrelated to source gloss sequences. This is another phenomenon that reflects a lack of faithfulness.

Based on the above observations, we conclude that enhancing the capability of SLT models to accurately identify visual signals is crucial, which can improve the faithfulness of SLT models. Besides, we assume imprecise gloss representations may hinder the further development of SLT models, and it is essential to increase the utilization of visual information. Different from recent works [12, 19] that explore the use of linguistic information to guide the learning of visual features, we prefer to take advantage of both modalities based on their different characteristics.

### 3.3 Improving Faithfulness in SLT

Previous works have shown remarkable success in modeling multi-lingual languages [37, 38] and cross-modal information [39] with a single transformer-based model [28], which verifies the capability of transformer-based models for aligning multi-source domains. Different from exploiting large-scale parallel corpora in a self-supervised way, we focus on how to make full use of existing supervised data in the low-resource setting. The proposed method includes a joint training scheme and two constraints.

**Joint Learning of SLT Subtasks.** All of the SLT subtasks (V2T, C2T, and G2T) are monotonically aligned and this characteristic of SLT indicates the acquired knowledge about translation can be shared across subtasks, which can not only control model complexity but also reduce overfitting. Therefore, we first integrate SLT subtasks into a single framework and learn them jointly. We adopt the pretrained word embedding of mBart [38] as previous work [15] to obtain the linguistic embedding sequence $\mathcal{E}(\mathcal{G})$ from a given gloss sequence $\mathcal{G}$. To bridge the gap between visual and linguistic modalities, we use a two-layer MLP $\mathcal{P}$ to obtain the visual embedding sequences $\mathcal{P}(\mathcal{V})$ and $\mathcal{P}(\mathcal{C})$. Besides, we share $\mathcal{P}$ and SLR classifiers for $\mathcal{V}$ and $\mathcal{C}$ to ensure the alignment between different kinds of visual features [40]. All of $\mathcal{E}(\mathcal{G})$, $\mathcal{P}(\mathcal{V})$, and $\mathcal{P}(\mathcal{C})$ are sent to the same translation module, and auxiliary translation losses are applied to the outputs of $\mathcal{P}(\mathcal{V})$ and $\mathcal{E}(\mathcal{G})$ for joint learning:

$$L_{joint} = L_{basic} + \lambda_{\mathcal{G}} L_{SLT}^{\mathcal{G}} + \lambda_{\mathcal{V}} L_{SLT}^{\mathcal{V}}, \tag{2}$$

where $\lambda_{\mathcal{G}}$ and $\lambda_{\mathcal{V}}$ are hyperparameters to control the balance among subtasks.

**Alignment Constraint.** The joint learning scheme shares the translation module among subtasks, but it is hard to identify the relationships between multiple subtasks and we try to further simplify this scheme. Code-switching [41] is a phenomenon that the alternation of languages within a conversation or utterance, which occurs when speakers are multilingual and familiar with correspondences among languages. As mentioned in Fig. 1(b), the source features of SLT subtasks are monotonically aligned, which provides a sufficient basis to let the SLT learner train with a multilingual learner jointly and make the SLT learner aware of word alignment implicitly. As shown in Fig. 2, we only keep two branches for SLT: the primary branch is training for the C2T subtask, and the auxiliary branch needs to tackle code-switching translation.

To generate a synthetic code-switching corpus for the auxiliary branch, we first estimate the alignment path $\hat{\pi} = \arg\max_{\pi} p(\pi|\mathcal{X}, \mathcal{G})$ with the maximal probability [42] from the recognition prediction of the primary branch, and then obtain the corresponding frames indexes $\eta_i$ from $\hat{\pi}$ for each gloss $i$. The code-switched sentence embedding $\mathcal{CS}(\mathcal{V}, \mathcal{G})$ is generated by replacing visual embeddings of each gloss in $\mathcal{P}(\mathcal{V})$ with the corresponding gloss embeddings $\mathcal{E}(\mathcal{G})$ (e.g., replacing $\mathcal{P}(\mathcal{V})_{\eta_i}$ with $\mathcal{E}(\mathcal{G}_i)$ for gloss $i$) with a probability of $\beta$:

$$\mathcal{CS}(\mathcal{V}, \mathcal{G}) = \mathrm{diag}\left(\mathbf{1} - \boldsymbol{m}(\beta)\right)\mathcal{P}(\mathcal{V}) + \mathrm{diag}(\boldsymbol{m}(\beta))\mathcal{E}(\mathcal{G}), \tag{3}$$

where $\boldsymbol{m}(\beta)$ is the mask vector for replacing and $\mathrm{diag}(\cdot)$ convert a vector to the corresponding diagonal matirx. In addition to the above gloss-wise code-switching, we also propose a sentence-wise generation process, which simply mixes embedding sequences as Mixup [43]:

$$\mathcal{CS}(\mathcal{V}, \mathcal{G}) = (1 - \beta)\mathcal{P}(\mathcal{V}) + \beta\mathcal{E}(\mathcal{G}). \tag{4}$$

It is worth noting that $\beta$ controls the ratio of gloss embeddings in the code-switched sentence, we adopt a larger $\beta$ at the beginning to leverage the fast convergence of the gloss embedding and then

gradually decay. To balance all subtasks and prevent overfitting, we further adopt a cyclical annealing schedule [44], which gradually reduces $\beta$ within each cycle:

$$\beta = \max(0, 1 - 2 * \mathrm{mod}(t - 1, M)/M), \tag{5}$$

where $t$ is the epoch number, and $M$ is the number of epochs for each cycle. We adopt a hyperparameter $\lambda_{\mathcal{CS}}$ to weight the auxiliary translation loss and formulate the total process as:

$$L_{align} = L_{basic} + \lambda_{\mathcal{CS}} L_{SLT}^{\mathcal{CS}}. \tag{6}$$

**Consistency Constraint.** The alignment constraint implicitly aligns visual and linguistic embeddings by sharing the translation module and leveraging synthetic code-switching corpora. However, there is a certain degree of complementarity between different kinds of subtasks: G2T takes discrete gloss embedding as input, which can easily capture correspondences between source and target languages but may lose detailed visual information, while V2T and C2T take continuous embedding as input, which contains more useful information about the sign but struggles to converge. To better leverage the characteristics of different subtasks and balance the training processes, we further propose a consistency constraint to regularize the SLT predictions between two branches:

$$L_{consist} = D_{KL}(p_{\mathcal{C}}||p_{\mathcal{CS}}) + D_{KL}(p_{\mathcal{CS}}||p_{\mathcal{C}}) \tag{7}$$

where $p_{\mathcal{C}}$ and $p_{\mathcal{CS}}$ are the predicted distribution over words based on features $\mathcal{C}$ and $\mathcal{CS}$, respectively, and $D_{KL}(\cdot, \cdot)$ denotes Kullback-Leibler divergence. When applying both constraints, the consistency constraint encourages the C2T model to find correct correspondences at the beginning of each cycle and gradually improves the importance of visual information as $\beta$ decays. The consistency constraint can also be explained from mutual learning [45] and learning from noisy labels [46].

Since the proposed method is based on the **Mono**tonically aligned nature of **SLT** subtasks, we named it **MonoSLT** and its final objective function is:

$$L_{final} = L_{align} + \lambda_c L_{consist}, \tag{8}$$

where $\lambda_c$ is the hyperparameter to balance constraints.

## 4 Experiments

### 4.1 Datasets and Evalution Metrics

**Datasets.** We evaluate MonoSLT on RWTHPHOENIX-Weather 2014T (Phoenix14T) and CSL-Daily datasets, and both datasets provide gloss and translation annotations.

◇ **Phoenix14T** [6] is an extension of the previous SLR dataset [3] by redefining segmentation boundaries and providing parallel gloss annotation and German translation. It is collected from weather forecast broadcasts and manually annotated, which indicates the gloss annotations may be imprecise. It has 8,257 sentences signed by 9 signers with vocabularies of around 1k glosses and 3k German words. There are 7096, 519, and 642 samples in training, dev, and test sets.

◇ **CSL-Daily** [11] is a Chinese sign language dataset with vocabularies of around 2k glosses and 2.3k Chinese characters. Different from Phoenix14T, CSL-Daily is collected by first designing the sign language corpus based on Chinese Sign Language textbooks and some Chinese corpora, and then inviting 10 signers to sign reference texts, which indicates the gloss annotations are quite precise. There are 18401, 1077, and 1176 samples in training, dev, and test sets.

**Evalution Metrics.** Similar to machine translation, BLEU [47] and ROUGH [48] scores (higher is better) are used to measure translation performance. We also report word error rate (WER, lower is better) to reflect the performance of SLR modules as previous works [3, 9, 15] do.

### 4.2 Implementation Details

For efficiency, we utilize MMPose [49] to estimate keypoint sequences from sign videos, and it generates 133 2D keypoints for each frame. We select 77 keypoints and divided them into five groups: 9 for body, 21 for each hand, 8 for mouth, and 18 for face. Group-wise modified ST-GCN [50] blocks are adopted to extract features from each group, and extracted features are projected to a vector of 1024 dimensions for each frame. For Conv1D, we adopt a 'C3-P2-C3-P2' structure, where C and P

Table 1: Performance comparison (%) on Phoenix14T dataset. The highest performance is highlighted in **bold**, while the second is underlined. ‡ denotes methods without using gloss annotations. † denotes methods only taking skeleton sequences as input. (R and B denote ROUGE and BLEU.)

| Sign2Text | Dev | | | Test | | | | | |
|---|---|---|---|---|---|---|---|---|---|
| | R | B4 | WER | R | B1 | B2 | B3 | B4 | WER |
| SL-Luong‡ [6] | 31.80 | 9.94 | - | 31.80 | 32.24 | 19.03 | 12.83 | 8.58 | - |
| TSPNet‡ [23] | - | - | - | 34.96 | 36.10 | 23.12 | 16.88 | 13.41 | - |
| JointSLRT [22] | - | 22.38 | 24.98 | - | 46.61 | 33.73 | 26.19 | 21.32 | 26.16 |
| STMC-T [14] | 48.24 | 24.09 | 21.1 | 46.65 | 46.98 | 36.09 | 28.70 | 23.65 | 20.7 |
| SignBT [11] | 50.29 | 24.45 | 22.7 | 49.54 | 50.80 | 37.75 | 29.72 | 24.32 | 23.9 |
| MMTLB [12] | 53.10 | 27.61 | 21.90 | 52.65 | 53.97 | 41.75 | 33.84 | 28.39 | 22.45 |
| SLTUNet [19] | 52.23 | 27.87 | 19.24 | 52.11 | 52.92 | 41.76 | 33.99 | 28.47 | - |
| TwoStream-SLT-K† [15] | 53.21 | 27.83 | 27.14 | 52.87 | 53.58 | 41.78 | 33.60 | 27.98 | 27.19 |
| TwoStream-SLT [15] | 54.08 | 28.66 | 17.72 | 53.48 | 54.90 | 42.43 | 34.46 | 28.95 | **19.32** |
| Baseline† | 53.22 | 27.55 | 21.5 | 52.56 | 53.69 | 40.96 | 32.84 | 27.37 | 21.1 |
| MonoSLT† | 55.41 | 29.96 | 21.2 | **55.73** | **57.05** | **44.70** | **36.73** | **31.15** | 21.4 |

Table 2: Performance comparison[4] on CSL-Daily dataset. The highest performance is highlighted in **bold**, while the second is underlined. * denotes methods with the inconsistent punctuation bug. The results of [6, 22] are reproduced by SignBT [11]. (R and B denote ROUGE and BLEU.)

| Sign2Text | Dev | | | Test | | | | | |
|---|---|---|---|---|---|---|---|---|---|
| | R | B4 | WER | R | B1 | B2 | B3 | B4 | WER |
| MMTLB* [12] | 53.38 | 24.42 | - | 53.25 | 53.31 | 40.41 | 30.87 | 23.92 | - |
| TwoStream-SLT* [15] | 55.1 | 25.76 | 25.4 | **55.72** | **55.44** | **42.59** | **32.87** | **25.79** | **25.3** |
| Baseline* | 50.85 | 22.83 | 29.1 | 50.96 | 52.11 | 38.97 | 29.46 | 22.74 | 28.2 |
| MonoSLT* | 52.58 | 23.67 | 29.1 | 52.58 | 52.65 | 39.72 | 30.27 | 23.53 | 28.2 |
| SL-Luong [6] | 34.28 | 7.96 | - | 34.54 | 34.16 | 19.47 | 11.84 | 7.56 | - |
| Joint-SLRT [22] | 27.06 | 11.88 | - | 36.74 | 37.38 | 24.36 | 16.55 | 11.79 | - |
| Sign-BT [11] | 49.49 | 20.8 | 33.2 | 49.31 | 51.42 | 37.26 | 27.76 | 21.34 | 32.2 |
| SLTUNet [19] | 53.58 | 23.99 | - | 54.08 | 54.98 | 41.44 | 31.84 | 25.01 | - |
| Baseline | 53.47 | 25.90 | 29.1 | 53.71 | 55.30 | 41.91 | 32.56 | 25.91 | **28.2** |
| MonoSLT | 55.28 | 26.91 | 29.1 | **55.35** | **55.87** | **42.75** | **33.52** | **26.83** | **28.2** |

denote 1D-CNN and max-pooling layer, respectively. Following [12], we utilize the official release of mBART-large-cc25 [2], which is pretrained on CC25 [3], as the initialization of the translation module. The default setting for hyperparameters: $\lambda_{\mathcal{C}}, \lambda_{\mathcal{G}}, \lambda_{\mathcal{V}}$ are set to 1.0 and $\lambda_c$ is set to 0.1 for simplicity. The beam width for the CTC decoder and the SLT decoder are 10 and 4, respectively. We train each model for 80 epochs with the cosine annealing schedule and an Adam optimizer, and the initial learning rate for each module: 1e-3 for the MLP, 1e-5 for the translation module, and 3e-3 for others. Each experiment is conducted on a single NVIDIA GeForce RTX 3090 GPU. Other details can be found in the supplementary.

## 4.3 Comparison with State-of-the-art

**Quantitative Comparison.** We report the performance of our MonoSLT model and relevant methods on Phoenix14T in Table 1. Because this paper mainly focuses on improving faithfulness, we put results of the Sign2Gloss2Text task in the supplementary. As shown in Table 1, we adopt a strong baseline, and the proposed method can bring further improvement (+3.79 BLEU-4). Besides, the proposed MonoSLT is not the best SLR approach, but outperforms the previous SLT method [15] with the best SLR performance by 2.2% (WER: 21.4% *vs.* 19.3%, BLEU-4: **31.15**% *vs.* 28.95%). MonoSLT also surpasses other previous methods with similar SLR performance, which indicates MonoSLT can increase the utilization of visual signals. This observation also reveals the lack of faithfulness in recent SLT methods, e.g., TwoStream-SLT [15] with multi-modality inputs (both skeleton sequence and video) achieve much better SLR performance than with skeleton sequence only (WER: 27.14% *vs.* 17.72%), but it achieves comparable SLT performance (BLEU-4: 28.23%

---

[2]https://huggingface.co/facebook/mbart-large-cc25

[3]https://commoncrawl.org/

[4]Our translation module is based on MMTLB (https://github.com/FangyunWei/SLRT), and we find it has an inconsistent punctuation bug during tokenization. For a fair comparison, we report results under both settings.

Table 3: A translation example of the lack of visual faithfulness on Phoenix14T dev set. We highlight the hallucination in red, and its corresponding correct translation and gloss in blue.

| | |
|---|---|
| SLR Ref: | morgen / sonne / ueberall / kueste / region / wolke / moeglich / regen neg-viel
( tomorrow / sun / everywhere / coast / region / cloud / possible / rain ) |
| SLR Hyp: | morgen / sonne / himmel (sky) / kueste / region / wolke / moeglich / regen neg-viel |
| SLT Ref: | am mittwoch im süden und an den küsten etwas regen sonst ist es meist freundlich .
(on wednesday in the south deland on the coasts some rain otherwise it is mostly friendly .) |
| Baseline Hyp: | am mittwoch im süden und nordosten hier und da regen sonst zum teil freundlich .
(on wednesday in the south and northeast here and there rain otherwise partly friendly .) |
| MonoSLT Hyp: | am mittwoch im süden und an den küsten etwas regen sonst ist es recht freundlich .
(on wednesday in the south and on the coasts some rain otherwise it is quite friendly .) |

Table 4: Ablation results (BLEU-4, %) of joint learning of SLT subtasks on Phoenix14T.

| Loss Weights | | | V2T | | C2T | | G2T | |
|---|---|---|---|---|---|---|---|---|
| $\lambda_\mathcal{V}$ | $\lambda_\mathcal{C}$ | $\lambda_\mathcal{G}$ | Dev | Test | Dev | Test | Dev | Test |
| 1.0 | - | - | 22.58 | 22.59 | - | - | - | - |
| - | 1.0 | - | - | - | 28.00 | 28.53 | - | - |
| - | - | 1.0 | - | - | - | - | 28.05 | 26.36 |
| 1.0 | 1.0 | - | **28.30** | 28.18 | **28.87** | 29.73 | - | - |
| - | 1.0 | 1.0 | - | - | 28.82 | 27.67 | **28.19** | 27.38 |
| 1.0 | 1.0 | 1.0 | 27.11 | 27.85 | 28.03 | 27.66 | 27.42 | 26.98 |

*vs.*28.95%) under these two settings. Besides, the proposed method improves faithfulness through joint learning and two constraints, which can be applied to any SLR model and has the potential to achieve better translation performance with a more powerful SLR model.

To show the generalization of the proposed method, we also report relevant performance on CSL-Daily in Table 2. As mentioned in Sect. 4.1, the CSL-Daily dataset has more precise gloss annotations, which indicates models with lower SLR performance can often achieve better SLT results. The proposed MonoSLT achieves inferior SLR and SLT performance than previous works [12, 15] but is still better than other works. Besides, the proposed method achieves better SLT performance than the baseline, which indicates that although glosses are precise intermediate tokenization, the lack of faithfulness still exists.

**Qualitative Comparison.** To provide a more intuitive understanding of the proposed method, we present a translation example in Table 3. It can be observed that part of the translation cannot find corresponding glosses, which indicates glosses are imprecise representations. Besides, the baseline generates hallucination 'and northeast here and there', while the corresponding gloss 'kueste' is correctly recognized but ignored by the translation module. The proposed MonoSLT can improve faithfulness and translate 'on the coasts' correctly. More results can be found in the supplementary.

### 4.4 Ablation and Discussion

**Ablation on Joint Learning.** As we mentioned in Sect. 3.3, the acquired knowledge about translation can be shared across SLT subtasks. We first evaluate different combinations of subtasks and present results in Table. 4. We notice that learning V2T and C2T subtasks jointly obtain the most significant improvements (5.72% for V2T and 0.87% for C2T), which shows V2T can achieve comparable results to G2T with proper regularization and the performance of V2T and C2T can be mutually improved. This observation also reveals a clear difference between the SLTUNet [19] and the proposed method: we pay more attention to the association between visual translation subtasks. Moreover, simply sharing more subtasks can not bring further improvements, which indicates the importance of designing proper solutions to exploit SLT subtasks.

**Ablation on Design Choices of MonoSLT.** To investigate the effectiveness of the proposed method, we present the ablation results of each design in Table 5. Both token-wise and sentence-wise code-switching achieve better performance than the best performance of joint learning, and we notice they can also accelerate training process and increase training stability. Adopting the cyclical annealing schedule can improve the G2T performance at the cost of a little performance loss of V2T and S2T, and combining it with the consistency constraint can bring further improvement. Besides, we can also observe that the proposed method can also improve the performance of G2T, which indicates S2T is also beneficial for G2T, and the previous 'G2T first' paradigm not fully exploits the potential of visual information.

Table 5: Ablation results (BLEU-4, %) of design choices of MonoSLT on Phoenix14T.

| Traning Scheme | Annealing | Consistency | V2T | | C2T | | G2T | |
|---|---|---|---|---|---|---|---|---|
| | | | Dev | Test | Dev | Test | Dev | Test |
| Joint Learning | | | 28.30 | 28.18 | 28.87 | 29.73 | 28.19 | 27.38 |
| Sentence-wise Code-switching | | | 28.35 | 29.13 | 29.20 | 29.07 | 26.92 | 26.30 |
| | ✓ | | 27.42 | 28.88 | 28.73 | 29.37 | 28.98 | 28.41 |
| | ✓ | ✓ | **28.91** | 30.18 | **29.69** | 30.76 | **29.67** | 28.08 |
| Token-wise Code-switching | | | 27.12 | 28.44 | 28.90 | 29.17 | 26.54 | 26.10 |
| | ✓ | | 28.42 | 29.14 | 28.77 | 29.87 | 29.26 | 29.71 |
| | ✓ | ✓ | **29.73** | 30.03 | **29.96** | 31.15 | **30.39** | 30.20 |

Table 6: Ablation results (BLEU-4, %) of source features and frozen layers on Phoenix14T.

| Source Feature | | Frozen Layer | | V2T | | C2T | | G2T | |
|---|---|---|---|---|---|---|---|---|---|
| Features | Logits | GCN module | Conv1D | Dev | Test | Dev | Test | Dev | Test |
| ✓ | | | | **29.73** | 30.03 | **29.96** | 31.15 | **30.39** | 30.20 |
| | ✓ | | | 28.26 | 29.55 | 28.99 | 29.77 | 28.08 | 27.55 |
| ✓ | | ✓ | | 28.59 | 29.73 | 29.20 | 30.21 | 28.89 | 29.66 |
| ✓ | | ✓ | ✓ | 29.29 | 31.45 | 29.68 | 31.21 | 30.06 | 30.98 |

**Ablation on Other Designs.** Compared to visual features, logits are a closer representation of glosses. As shown in Table 6, adopting logits as input leads to a little performance degradation, which also indicates glosses are imprecise representations of signs on Phoenix14T. To further explore the origin of performance gain, we evaluate the effects of frozen modules in Table 6. Frozing both the GCN-based module and Conv1D can achieve comparable results, which indicates the improvement mainly comes from making better use of existing features, rather than extracting new visual features. Adopting the frozen version of MonoSLT can also improve training efficiency.

**Limitations and Discussions.** Although the proposed MonoSLT achieves competitive results on two benchmarks, we notice several limitations of our model. First, the proposed method is motivated to solve the hallucination problem of SLT, however, we have not found proper metrics to quantitatively evaluate the faithfulness of SLT models and still use BLEU and ROUGE for evaluation. We believe faithfulness is important when SLT is applied in real life because an unfaithful SLT model may produce unexpected consequences. Secondly, although the proposed method can improve faithfulness in SLT, as shown in Table. 6, it does not extract new visual features, which indicates that expensive gloss annotations are still essential. Designing effective gloss-free is a fascinating route for SLT. Third, although we design several approaches to make the training stable, it still encounters difficulties in converging occasionally, and we will continue to enhance its stability.

## 5 Conclusion

Faithfulness is one of the desired criteria to evaluate the applicability of SLT models. In this paper, we explore the association among different SLT-relevant tasks and reveal that the lack of faithfulness exists in recent SLT methods. To improve faithfulness in SLT, we attempt to increase the utilization of visual signals in SLT and propose a framework named MonoSLT, which leverages the monotonically aligned nature of SLT subtasks to train them jointly. We further propose two kinds of constraints to align visual and linguistic embeddings and leverage the advantage of subtasks. Experimental results show that the proposed MonoSLT is competitive against previous SLT methods by increasing the utilization of visual signals, especially when glosses are imprecise. We hope the proposed method and empirical conclusions can inspire future studies on SLT and relevant tasks.

**Broader Impact** This paper focuses on improving faithfulness in SLT to bridge the communication gap between the Deaf and hearing communities. Although the MonoSLT has made some progress there still seems a long way to go. Please note this research is limited to public datasets which have limited samples and are collected under constrained conditions, and the findings may not directly transfer to other scenarios or domains. Domain expertise and human supervision are essential when using it to make critical decisions, which may generate erroneous or potentially harmful translations. Moreover, potential biases in the training data or method may introduce limitations or assumptions that need to be considered when using it.

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
