# A  Appendix

## A.1  Implementation Details

As present in Sect. 4.2, we adopt MMPose[1] to estimate keypoints from each frame, specifically, we choose the HRNet [1] trained on the COCO-WholeBody v1.0 [2]. We select 77 keypoints from 133 estimated COCO format 2D keypoints, the selected keypoints are visualized in Fig. 1. The GCN-based module is modified based on ST-GCN[2], and we adopt three modified ST-GCN blocks for each group of keypoints, and the output dimensions of them are 64, 128 and 256, respectively. For Conv1D, we adopt a 'C3-P2-C3-P2' structure, where $C\alpha$ and $P\beta$ denote temporal convolutional layer with the kernel size of $\alpha$ and 1D pooling layer with the kernel size of $\beta$, which reduces the sequence length from $T$ to $T/4$. The output dimension of the BiLSTM layer is 1024 (512 for each direction). Both the weights and the fed features of the shared classifiers are normalized.

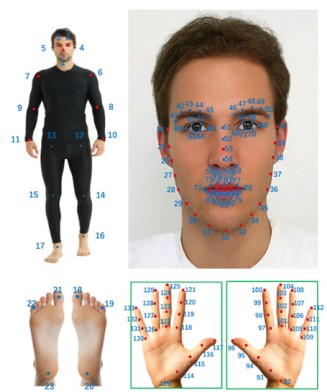

Figure 1: The 133 keypoints of the estimation output. The selected 77 keypoints are marked in red.

We train the SLR models for 40 epochs with an Adam optimizer. The initial learning rate is 4e-4 and divided by 10 at epochs 20 and 35. We adopt a weight decay of rate 0.001. We adopt the pretrained SLR model as the initialization of the training of SLT models for efficiency. We train the SLT models on CSL-Daily for 40 epochs, and 80 for Phoenix14T. Because the gloss annotations of CSL-Daily are precise and we gloss probability as input for the MLP, and adopt a one-layer MLP initialized using the weights of the pretrained gloss embeddings as previous work does [3]. The implementation of SLR model[3] is based on VAC [4] and the implementation of SLT model[4] is based on MMTLB [3]. For the cyclical annealing schedule, we set $M$ to 20 for the number of epochs of each cycle. For the joint learning of multiple subtasks, we share both the MLP layers and the translation module between V2T and C2T tasks, and further share the translation module with G2T. The only augmentation used in this paper is the random temporal rescale ($\pm20\%$).

The mentioned inconsistent punctuation in Sect. 4.4 is produced because the tokenizer takes the Chinese punctuation but generates English punctuation after decoding, while the ground truth is the Chinese punctuation, which leads to a large performance degradation.

---

[1] https://github.com/open-mmlab/mmpose/blob/dev-1.x/configs/wholebody_2d_keypoint/topdown_heatmap/coco-wholebody/td-hm_hrnet-w48_8xb32-210e_coco-wholebody-384x288.py

[2] https://github.com/yysijie/st-gcn

[3] https://github.com/ycmin95/VAC_CSLR

[4] https://github.com/FangyunWei/SLRT/tree/main/TwoStreamNetwork

Table 1: Performance comparison (%) on Phoenix14T dataset. The highest performance is highlighted in **bold**, while the second is underlined. ‡ denotes methods without using gloss annotations. * denotes evaluating with code-switching. (R and B denote ROUGE and BLEU.)

| Sign2Gloss2Text | Dev | | | Test | | | | | |
|---|---|---|---|---|---|---|---|---|---|
| | R | B4 | WER | R | B1 | B2 | B3 | B4 | WER |
| SL-Luong‡ [5] | 44.14 | 18.40 | - | 43.80 | 43.29 | 30.39 | 22.82 | 18.13 | - |
| JointSLRT [6] | - | 22.11 | 24.98 | - | 48.47 | 35.35 | 27.57 | 22.45 | 26.16 |
| SignBT [7] | 49.53 | 23.51 | 22.7 | 49.35 | 48.55 | 36.13 | 28.47 | 23.51 | 23.9 |
| STMC-T [8] | 46.31 | 22.47 | 21.1 | 46.77 | 48.73 | 36.53 | 29.03 | 24.00 | 20.7 |
| MMTLB [9] | 50.23 | 24.63 | 21.90 | 49.59 | 49.94 | 37.28 | 29.67 | 24.60 | 22.45 |
| SLTUNet [10] | 49.61 | 25.36 | 19.24 | 49.98 | 50.42 | 39.24 | 31.41 | 26.00 | - |
| TwoStream-SLT [3] | 52.01 | 26.47 | 27.14 | 51.59 | 52.11 | 39.81 | 32.00 | 26.71 | **19.32** |
| Baseline | 51.12 | 25.93 | 21.5 | 51.52 | 52.52 | 39.99 | 32.28 | 27.03 | 21.1 |
| MonoSLT | 44.09 | 20.92 | 21.2 | 44.18 | 45.54 | 32.92 | 25.49 | 20.81 | 21.4 |
| MonoSLT* | 54.40 | 28.90 | 21.2 | **54.38** | **55.53** | **43.00** | **34.87** | **29.25** | 21.4 |

## A.2 Sign2Gloss2Text Results

In this paper, we focus on improving faithfulness in SLT by sharing the acquired knowledge among SLT subtasks, rather than improving the SLR or G2T task. As mentioned in Sect. 4.3, we present S2G2T results on Phoenix14T in Table 1. Because we adopt a shared translation module among SLT subtasks, and train V2T and G2T subtasks in a code-switching way, which means we do not train independent G2T actually. As shown in Table 1, the proposed MonoSLT achieves awful performance when masking visual embeddings in Equ. 3, which can be formulated as:

$$\mathcal{S}(\mathcal{V}, \mathcal{G}) = \operatorname{diag}(m)\mathcal{E}(\hat{\mathcal{G}}), \tag{1}$$

where $m$ is the mask vector that masking blank frames in $\hat{\pi} = \arg\max_\pi p(\pi|\mathcal{X}, \hat{\mathcal{G}})$, and $\hat{\mathcal{G}}$ denotes the recogized gloss sequence from the SLR model. However, the performance can be significantly improved by 'unmasking' the visual embeddings of non-blank frames (corresponding to MonoSLT* in Table 1):

$$\mathcal{S}(\mathcal{V}, \mathcal{G}) = (\mathbf{1} - \operatorname{diag}(m))\mathcal{P}(\mathcal{V}) + \operatorname{diag}(m)\mathcal{E}(\hat{\mathcal{G}}). \tag{2}$$

These results indicate that it is not suitable to compare the S2G2T performance of the proposed MonoSLT with others, which may lead to unfair comparison. Therefore, we only provide it in this supplementary for reference.

## A.3 More Qualitative Results

In Sect. 4.4, we present a translation example to provide a more intuitive understanding of the proposed method, and we provide more examples in Table 2. It can be observed that sharing the acquired knowledge among the SLT subtasks can **reduce the accumulated errors** (the SLR model fails to recognize some gloss and the baseline solution generates corresponding wrong translations

in the first three samples, while MonoSLT can predict the correct translation based on visual information) and **improve the faithfulness** (the SLR model recognizes all glosses but the baseline solution generates hallucinations in the last three samples, while the MonoSLT can generate more faithful translations).

Table 2: More translation examples of the lack of visual faithfulness on Phoenix14T dev set. We highlight the hallucination in red, and its corresponding correct translation and gloss in blue.

| | |
|---|---|
| SLR Ref: | jetzt / wetter / wie-aussehen / morgen / mittwoch / eins / dreissig / maerz
( now / weather / how to look / tomorrow / wednesday / one / thirty / march ) |
| SLR Hyp: | ***** / wetter / wie-aussehen / morgen / mittwoch / eins / zwanzig (twenty) / maerz |
| SLT Ref: | und nun die wettervorhersage für morgen mittwoch den einunddreiSSigsten märz .
(and now the weather forecast for tomorrow wednesday thirty-first of march .) |
| Baseline Hyp: | und nun die wettervorhersage für morgen mittwoch den einundzwanzigsten märz .
(and now the weather forecast for tomorrow wednesday twenty-first of march .) |
| MonoSLT Hyp: | und nun die wettervorhersage für morgen mittwoch den einunddreiSSigsten märz .
(and now the weather forecast for tomorrow wednesday thirty-first of march .) |
| SLR Ref: | jetzt / wetter / morgen / freitag / sieben / zwanzig / august
( now / weather / tomorrow / Friday / seven / twenty / August ) |
| SLR Hyp: | jetzt / wetter / morgen / dienstag (Tuesday) / sieben / zwanzig / august |
| SLT Ref: | und nun die wettervorhersage für morgen freitag den siebenundzwanzigsten august.
(and now the weather forecast for tomorrow friday twenty seventh august.) |
| Baseline Hyp: | und nun die wettervorhersage für morgen dienstag den siebenundzwanzigsten august.
(and now the weather forecast for tomorrow tuesday the twenty seventh of august.) |
| MonoSLT Hyp: | und nun die wettervorhersage für morgen freitag den siebenundzwanzigsten august.
(and now the weather forecast for tomorrow friday the twenty seventh of august.) |
| SLR Ref: | **** / ******* / koeln / region / ix / dreizehn / grad / alpen / drei / grad
( **** / ********/ coeln / region / ix / thirteen / degrees / alpes / three / degrees ) |
| SLR Hyp: | nord / deutsch (German) / land / region / ix / ******** / grad / berg (moutain) / frei (free) / grad |
| SLT Ref: | in der kölner bucht heute nacht milde dreizehn am alpenrand drei grad .
(in the cologne bay tonight mild thirteen at the alpine rim three degrees.) |
| Baseline Hyp: | in ganz deutschland am niederrhein bis dreizehn am alpenrand bis dreiSSig grad .
(all over germany on the lower rhine up to thirteen degrees on the alpine rim up to thirty degrees .) |
| MonoSLT Hyp: | in der kölner bucht heute nacht plus drei am alpenrand bis plus drei grad .
(in the cologne bay tonight plus three at the alpine rim up to plus three degrees .) |
| SLR Ref: | auch / samstag / nächste / gleich / ix / kommen / mehr / kuehl
( also / saturday / next / coming soon / ix / come / more / cool ) |
| SLR Hyp: | auch / samstag / in-kommend (incoming) / gleich / ix / kommen / mehr / kuehl |
| SLT Ref: | ähnliches wetter auch am samstag von norden wird es allerdings kühler .
(similar weather also on saturday delfrom the north, but it will be cooler.) |
| Baseline Hyp: | auch am samstag ändert sich an diesem wetter wenig von norden flieSSt kühlere luft heran .
(also on saturday little changes in this weather cooler air flows in from the north .) |
| MonoSLT Hyp: | auch am samstag ähnliches wetter von norden wird es kühler .
(also on saturday similar weather from the north it gets cooler .) |
| SLR Ref: | morgen / tag / achtzehn / maximal / drei / zwanzig / grad
( tomorrow / day / eighteen / maximum / three / twenty / degree ) |
| SLR Hyp: | morgen / tag / achtzehn / maximal / drei / zwanzig / grad |
| SLT Ref: | am tag dann temperaturen zwischen achtzehn und dreiundzwanzig grad .
(during the day temperatures between eighteen and twenty-three degrees.) |
| Baseline Hyp: | am tag achtzehn bis dreiundzwanzig grad .
(during the day eighteen to twenty-three degrees .) |
| MonoSLT Hyp: | am tag werte zwischen achtzehn und dreiundzwanzig grad .
(during the day values between eighteen and twenty-three degrees .) |
| SLR Ref: | nord morgen koennen sonne
( north / tomorrow / can / sun ) |
| SLR Hyp: | nord morgen koennen sonne |
| SLT Ref: | im norden kann sich morgen auch mal die sonne blicken lassen .
(in the north, the sun may also make an appearance tomorrow.) |
| Baseline Hyp: | im nordosten zeigt sich morgen gelegentlich die sonne .
(in the northeast the sun will occasionally show tomorrow.) |
| MonoSLT Hyp: | im norden zeigt sich morgen auch mal die sonne .
(in the north the sun may also show itself tomorrow .) |