# OpenReview forum: "Towards Faithful Sign Language Translation"
_NeurIPS.cc/2023/Conference — Submitted to NeurIPS 2023_

### Official Review · Reviewer_6xPF · 2023-07-04

**Soundness:** 3 good
**Presentation:** 3 good
**Contribution:** 3 good
**Rating:** 6
**Confidence:** 5

**Summary:**

The paper proposes a novel framework for sign language translation (SLT), which can integrate multiple SLT subtasks. The work is motivated by a series of experimental analysis, e.g., converging speed of different subtasks and relationship between SLT and sign language recognition (SLR) performance. Besides, two constraints are proposed to improve the faithfulness of the model and ease the model training. The method achieves SOTA performance on two widely adopted SLT benchmarks using only keypoint inputs.

**Strengths:**

1. The paper figures out two important problems: the lack of faithfulness in current SLT models and the inconsistent trend between SLR and SLT, which can inspire future works in this field.
2. The method is well motivated by a series of in-depth analysis in Figure 3.
3. The code-switching operation is interesting and novel, which can inspire future works on cross-modality modeling for SLT.
4. SOTA performance are achieved with a lightweight model using only keypoint inputs.


**Weaknesses:**

My major concerns come from method details and experiments.

Method:

1. The MLP and classifier in Figure 2 should be shared or not? In VAC [9], two different classifiers are appended to the visual and contextual module, while SMKD [40] uses a shared classifier, and the paper follows the design of SMKD. Intuitively, different classifiers should be used to project two features from different spaces into a common space. More discussion is needed for the discrepancy.
2. Gloss embeddings and mixup also appeared in a recent paper in the field of SLR [R1]. In [R1], the gloss embeddings are extracted by FastText, and the mixup is also achieved between visual and gloss embeddings. Some discussion or comparison should be added.
3. What is the motivation to fulfill code switch between the visual and gloss embeddings? Is it possible to use it between the contextual and gloss embeddings?

Experiments:

4. In Table 5, the sentence-wise code-switching does not consistently outperform the token-wise counterpart. The authors may explain why the sentence-wise one performs better when not using annealing and consistency.
5. As stated in line 299, logits are a closer representation of glosses. Also, [12] uses logits as the input for the translation module for CSL-Daily. Thus, it is not rigorous to conclude that adopting logits will degrade the performance since the ablation study is conducted on Phoenix14T.
6. The paper focuses on improving the faithfulness of SLT. But there are not objective metrics mentioned to measure the faithfulness.

[R1] Natural Language-Assisted Sign Language Recognition, CVPR 2023.

**Questions:**

1. How to obtain $m(\beta)$? Is it right that: when $\beta>0.5$, $m(\beta)=1$?
2. What is the difference between the consistency constraints in the paper with that in VAC?

**Limitations:**

The authors have discussed the limitations and societal impact adequately.

---

> ### Author Rebuttal · Authors · 2023-08-08
>
> Thank you for your positive feedback and constructive comments. Our review responses are summarized below.
>
> **Q6. About the choice of evaluation metrics.**
>
> Thanks for your constructive comments on the evaluation metrics. As we mentioned **in the “global” response**, we suggest evaluating the over-translation and under-translation with BLEU-1 and ROUGE-1, respectively. We also evaluate the proposed method on the subsets of different universal part-of-speech tags. Experiment results show that almost all tags benefit a lot from the proposed MonoSLT, which indicates the proposed method is effective in increasing faithfulness.
>
> **Q1. Whether the MLP and classifier in Figure 2 should be shared or not?**
>
> Thanks for your insightful comments. This submission follows the design of SMKD and shares the MLP and classifiers. We agree that it is common to project two features from different spaces into a common space. However, the two feature spaces here are not independent. The contextual features are extracted from the visual features with a two-layer BiLSTM, thus sharing the classifiers and the following modules can be regarded as a kind of implicit constraint for the BiLSTM module. Besides, as we mentioned in the “global” response, this submission keeps the visual encoder (SMKD-based) the same and does not dig into the faithfulness of the visual encoder. The visual encoder already projects two features from different spaces into a common space, and sharing the following modules or not makes little influence on the performance as shown in Table 6.
>
> **Q2. Comparison with recent work.**
>
> Thanks for pointing out the relevant work and it has a similar motivation of alignment constraint. Both works try to guide the learning of visual features with relevant semantic meanings. The main difference here is the NLA-SLR conducts a class-wise mixup to improve isolated sign language recognition, and this submission conducts a sentence-wise mixup to align make the translation module aware of word alignment implicitly. We will add this comparison in the related work.
>
> **Q3. What is the motivation to fulfill code switch between the visual and gloss embeddings?**
>
> As mentioned in the introduction of Sect. 3.3, recent works have shown that a single transformer-based model has the capacity to model multi-lingual languages (e.g., XNLI[37], mBart[38]) and cross-modality information (e.g., VideoBERT[39]). If we regard the translation model as an auto-encoder, the token-wise code-switching can be regarded as an ensemble of two complementary masked language modeling processes, and they need contextual information from each other to achieve accurate translation. Therefore, the translation module needs to align the contextual information from both visual and semantic features to achieve the translation goal during training, which aligns the visual and semantic spaces implicitly. This submission adopts the code-switching module to ease the training of visual features and align the feature spaces, it is also possible to be used as an augmentation strategy to the "self-alignment" of visual or contextual features, which is verified in the experiments on CSL-Daily **in the "global" response**.
>
> **Q4. The choice of code-switching module.**
>
> This submission provides two kinds of code-switching approaches and both of them achieve comparable results as shown in Table 5. Compared to choosing one as the universal solution, we suggest selecting approaches based on the practical situation. For example, sentence-wise code-switching is a more suitable choice when the alignment is hard to obtain because it can preserve the complete information of both sequences, while token-wise code-switching may be a more suitable choice when the model is co-trained with a masked language modeling task.
>
> **Q5. Conclusion that adopting logits will degrade the performance is not rigorous.**
>
> Thank you for bringing this up. This submission conducts this experiment to reveal that the logits are imprecise on Phoenix14T. We apologize for the misunderstanding and will revise this statement as "adopting logits as input leads to a little performance degradation on Phoenix14T, ...".
>
> **Q7. How to obtain $m(\beta)$?**
>
> $m(\beta)$ is a vector of length $T$, and each element is $0$ or $1$, which represents whether the corresponding semantic embedding is used or not. Specifically, we first estimate the alignment path $\hat{\pi}=\arg\max_\pi p(\pi|X, G)$ with the maximal probability [41], and then obtain the corresponding range $[s_i, e_i)$ for each gloss $i$, where the $s_i$ and $e_i$ denote start and end frame indexes. Then we randomly sample gloss $i$ with the probability of $\beta$ and replace the corresponding visual embeddings $\mathcal{P}(v_i), i \in [s_i, e_i)$ with $\mathcal{E}(G_i)$ (i.e., replace all frames that are recognized as gloss $i$ with its semantic embedding). The matrix form is presented in Equ. 3. We will clarify the relevant formulation and provide several examples in the supplementary.
>
> **Q8. What is the difference between the consistency constraints in the paper with that in VAC?**
>
> The main difference is that VAC needs pair-wise alignments, and we select two subtasks based on their characteristic and incorporate them as a single task to simplify the training process. Therefore, from the perspective of constraint design, the proposed method can be regarded as a simplified version of VAC. We choose a VAC-style constraint design to validate the effectiveness of the proposed framework, and other alignment methods are also potentially feasible (e.g., contrastive-based methods).

---

### Official Review · Reviewer_yVLq · 2023-07-04

**Soundness:** 3 good
**Presentation:** 3 good
**Contribution:** 3 good
**Rating:** 5
**Confidence:** 4

**Summary:**

This paper mainly discusses the challenges of improving faithfulness in sign language translation and proposes solutions. The researchers leverage rich monolingual data and adopt back-translation to generate synthetic parallel data, explore the potential of denoising auto-encoder, and propose the MonoSLT framework to improve the accuracy of sign language translation. They also emphasize the importance of alignment and consistency constraints to align visual and linguistic embeddings and improve faithfulness. This paper has important reference value for improving faithfulness in sign language translation.

**Strengths:**

1. Proposing a new unified framework, MonoSLT, which integrates subtasks of sign language translation into a single framework, allowing these subtasks to share acquired knowledge.

2. Proposing two constraints: alignment constraint and consistency constraint, which help improve the faithfulness of translation.

3. Experimental results show that the MonoSLT framework is competitive in improving the faithfulness of sign language translation and can increase the utilization of visual signals, especially when sign language vocabulary is imprecise.

**Weaknesses:**

The paper does not explicitly address the handling of non-manual signals and sign language morphological changes, which are crucial factors influencing the faithfulness of sign language translation.

The experimental settings in the paper do not provide detailed explanations for many hyperparameters.

**Questions:**

Why is the overfitting problem in the C2T model defined as a lack of faithfulness?

Using regularization to address overfitting seems to make sense, but a more common approach should be a suitable variant of Dropout. This requires further explanation.

The paper sets many hyperparameters, but lacks corresponding ablation studies, as well as the analysis of the mutual influences between different parameters.

Can simply setting all hyperparameters to 1.0 or 0.1 yield better results than state-of-the-art (SOTA) methods? This question remains unanswered.

**Limitations:**

The proposed method lacks proper metrics to quantitatively evaluate the faithfulness of Sign Language Translation models and continues to use BLEU and ROUGE for evaluation.

The paper's analysis and experimental results regarding faithfulness are not clearly defined.

---

> ### Author Rebuttal · Authors · 2023-08-07
>
> Thanks for the insightful comments. Our responses to them are summarized below.
>
> **Q1. About the non-manual signals.**
>
> We agree with you that non-manual signals are essential for SLU. One of the major reasons that we select keypoints as input is that synchronous signals can be easily modeled by the interactions among different groups of keypoints. As mentioned in the supplementary, to make the skeleton-based SLR method achieves comparable performance with image-based SLR methods, we divide the keypoints into five groups (body, left hand, right hand, mouth, and facial contour), apply modified ST-GCN blocks to model each signal independently and model the interaction among groups through 1D CNN layers. To better reveal the influence of non-manual signals for SLT, we conduct an ablation study by masking the rest group-wise features.
>
> ||Dev Set|||Test Set|||
> |---|:---:|:---:|:---:|:---:|:---:|:---:|
> ||BLEU|ROUGE|WER|BLEU|ROUGE|WER|
> |Body|18.84|45.07|42.0|20.69|45.11|40.0|
> |Body+Hands|26.38|51.72|28.0|26.12|50.44|27.7|
> |Body+Hands+Mouth|28.27|53.93|21.6|29.06|53.70|22.0|
> |Body+Hands+Mouth+Face|29.96|55.41|21.2|31.15|57.05|21.4|
>
> As shown in the table above, the mouth signal is essential for SLU, and including the facial signal can bring further improvement. However,  we believe the leverage of visual signals is more relevant to the SLR task. Therefore, we prefer to add this ablation in the supplementary to make the completeness of the main body.
>
> **Q2. About the sign language morphological changes.**
>
> Thanks for your suggestion about the morphological changes, which lead to a large intra-class variance. As mentioned in the supplementary, we normalize the weights and the fed features of the shared classifiers as previous metric learning works do to reduce intra-class variance. We agree that modeling morphological changes of sign languages are important for faithfulness, as the sign gloss may appear different under different contexts. However, we believe that exploring the relationship among SLT subtasks has a higher priority than modeling morphological changes. As shown in the experiments of **the “global” response**, although we do not explicitly model morphological changes,  the alignment among SLT subtasks improves the faithfulness on ADJ, ADP, and PRON, which indicates that the proposed method can improve the ability to handle the morphological changes implicitly.
>
> **Q3. About the hyperparameters settings and why setting all hyperparameters to 1.0 or 0.1 yields better results.**
>
> Thanks for your suggestion about the hyperparameters settings. Most of the hyperparameters are set based on previous works, for example, the temperature is set to 1 as the original knowledge distillation. As for loss weights, we pay more attention to the loss designs and keep the loss weights the same as the preliminary experiment in Table 4. To provide more details, we provide more ablation results about the choice of the hyperparameters on Phoenix14T.
>
> It is worth noting that the following results are evaluated on the G2T task because we saved the best model on the dev set based on the performance of G2T by mistake, but the influence of hyperparameter choice on performance should be similar.
>
> Ablation on the temperature of KL divergence
> |Temperature|BLEU|ROUGE|BLEU|ROUGE|
> |:---:|:---:|:---:|:---:|:---:|
> |1|29.73|55.09|30.03|55.17|
> |2|**30.26**|55.43|29.65|54.36|
> |4|29.82|**55.57**|29.07|55.03|
> |8|29.85|55.49|29.35|54.42|
>
> Based on this ablation, the influence of temperature is little and setting it as 2 can achieve better performance.
>
> Ablation on the weight of the consistency loss
> ||Dev Set| |Test Set| |
> |:-:|:-:|:-:|:-:|:-:|
> |$\lambda_c$|BLEU|ROUGE|BLEU|ROUGE|
> |0.0|28.77|54.05|29.87|54.61|
> |0.01|29.36|54.30|27.96|52.65|
> |0.02|29.80|55.35|29.95|54.88|
> |0.05|29.62|55.25|28.71|53.52|
> |0.1|**30.26**|55.43|29.65|54.36|
> |0.2|30.08|**55.46**|29.82|54.57|
> |0.5|28.22|54.00|29.06|53.88|
> |1.0|12.57|36.45|12.95|36.35|
>
> In the main paper, we set $\lambda_c=0.1$ in default, because a large loss weight will domain the training process and affect the translation performance. It can be observed that $[0.02, 0.2]$ is a reasonable range for $\lambda_c$.
>
> We will add more ablation results about hyperparameter choices in the supplementary.
>
> **Q4. Why is the overfitting problem in the C2T model defined as a lack of faithfulness?**
>
> As shown in Fig. 3(b), the different convergence speeds make the C2T model easier to leverage the target-side context and the implicit language model rather than visual features, which faces a pretty high risk of hallucination. As suggested by reviewer uJKb, we agree that overfitting is caused by faithfulness is not well-supported, many possible factors lead to overfitting, and the lack of faithfulness is only one of them. The conclusion here should be an assumption that motivates us to explore the leverage of visual signals in SLT. We will carefully revise the relevant analysis to make it more thorough.
>
> **Q5. Regularization or Dropout?**
>
> Thanks for your comment, it provides another viewpoint for the proposed method. The code-switching module (Equ.3) can also be regarded as a joint training of two source-aligned translation tasks with complementary token-wise dropout. Designing regularization loss is a more familiar route for us and adopting dropout or other regularization tools is also a feasible choice.
>
> **Q6. About the evaluation metrics.**
>
> Thanks for your constructive comments on the evaluation metrics. As we mentioned **in the “global” response**, we suggest evaluating the over-translation and under-translation with BLEU-1 and ROUGE-1, respectively. We also evaluate the proposed method on the subsets of different universal part-of-speech tags. Experiment results show that almost all tags benefit a lot from the proposed MonoSLT, which indicates the proposed method is effective in increasing faithfulness.

---

> > ### Comment · Reviewer_yVLq · 2023-08-17
> >
> > The authors answered all of my questions very carefully and added more extensive experimental results. I hope they will continue to consider this deeply in future studies. The rating will be changed to 5.

---

### Official Review · Reviewer_ujKb · 2023-07-07

**Soundness:** 3 good
**Presentation:** 3 good
**Contribution:** 3 good
**Rating:** 4
**Confidence:** 4

**Summary:**

The paper discusses the issue of faithfulness in sign language translation (SLT), which refers to whether the SLT model captures the correct visual signals. It is found that imprecise glosses and limited corpora can hinder faithfulness in SLT. To address this, the paper proposes MonoSLT, which integrates SLT subtasks into a single framework that can share knowledge among subtasks. Two kinds of constraints are proposed to improve faithfulness: the alignment constraint and the consistency constraint. Experimental results show that MonoSLT is competitive against previous SLT methods and can increase the utilization of visual signals, especially when glosses are imprecise.

**Strengths:**

The method proposed in this paper outperforms multiple baseline methods, which is a promising contribution to sign language translation.

**Weaknesses:**

1. There is no comparison with [12,15] on the bug-free dataset, which is a concern. Although I understand that reproducing [15] would require additional effort, since your code is based on MMTLB, it would be reasonable to verify the effectiveness of MMTLAB on the bug-free data.

2. The analysis of faithfulness and hallucination in the paper is not in-depth enough. There is no metric (either manual or automatic) to quantify faithfulness and hallucinations, and the improvement in BLEU is not sufficient to indicate that the faithfulness issue has been effectively addressed. The few cases presented in the paper are not enough to support the conclusions.

3. The analysis in section 3.2 is not thorough enough, and the conclusions are somewhat forced. For example, the statement that overfitting is caused by faithfulness is not well-supported, and the conclusion that there is no obvious negative correlation between SLT and SLR in Figure 3(c) is due to hallucination lacks data support and quantitative analysis. The few examples presented in section 4 are not sufficient to demonstrate the issue of hallucination.

**Questions:**

n/a

**Limitations:**

see weakness

---

> ### Author Rebuttal · Authors · 2023-08-08
>
> **Q1. About the bug-free evaluation on the CSL-Daily dataset.**
>
> Thanks for your suggestion, it pushes us to figure out the performance gap between MMTLB[12] and the used baseline. We present the re-evaluated results **in the "global" response**. Moreover, we also find that the overconfident predictions of the BiLSTM layer hinder the leverage of visual information. Therefore, we only train V2T and G2T jointly on CSL-Daily. With this modification, the gap between MMTLB and the used baseline is narrowed, and MonoSLT shows competitive performance with skeleton inputs only. We will clarify relevant statements and update the results in Table 2.
>
> **Q2. About the evaluation metrics.**
>
> Thanks for your constructive comments on the evaluation metrics. As we mentioned **in the “global” response**, we suggest evaluating the over-translation and under-translation with BLEU-1 and ROUGE-1, respectively. We also evaluate the proposed method on the subsets of different universal part-of-speech tags. Experiment results show that almost all tags benefit a lot from the proposed MonoSLT, especially for ADP, ADV, and PRON, which indicates the proposed method is effective in increasing faithfulness.
>
> **Q3. The analysis in section 3.2 is not thorough enough, and the conclusions are somewhat forced.**
>
> Thanks for your constructive comments on the analysis in section 3.2. We agree with you that overfitting caused by faithfulness is not well-supported, many possible factors lead to overfitting, and faithfulness is only one of them. The conclusion here should be an assumption that motivates us to explore the leverage of visual signals in SLT. As for the hallucination, we agree that there is no obvious negative correlation between SLT and SLR in Figure 3(c) can only show the potential risk of hallucination, which is also a motivation rather than a conclusion. Based on the evaluation results (0.570 BLUE-1 in total) **in the "global" response**, about 43% of words predicted by MonoSLT do not exist in the references, although some of them perhaps are synonyms of the corresponding words in the reference, the MonoSLT model still faces a pretty high risk of hallucination. We will carefully revise the relevant analysis to make it more thorough.

---

### Official Review · Reviewer_mHke · 2023-07-07

**Soundness:** 2 fair
**Presentation:** 3 good
**Contribution:** 3 good
**Rating:** 5
**Confidence:** 5

**Summary:**

This work is dedicated to enabling the SLT model to capture correct visual signals (faithfulness in SLT). In order to improve faithfulness in SLT, the author integrates SLT subtasks into a single framework named MonoSLT, and based on this, proposes alignment constraints and consistency constraints. The former is used for aligning the visual and linguistic embeddings. The latter is used for integrating the advantages of subtasks. To demonstrate the effectiveness of the proposed method, the authors conduct experiments on two public datasets.

**Strengths:**

[1 - complete layout and detailed description]. The article has a relatively complete overall layout and a detailed work description.

[2 – method novelty]. The author Introduced the code-switching phenomenon in the Alignment Constraint, mimicking the phenomenon of language alternation in conversations between multilinguals, and mixed visual embedding and gloss embedding as an input to the Translation Module.

[3 – the rationality of alignment]. Implicitly align visual and linguistic embeddings through shared translation modules and synthetic code-switching corpora. Better utilization of the characteristics of different subtasks.

[4 – method performance]. On the Phoenix14T dataset, the author's method only uses skeleton sequences as input, which improves performance (+2.2 BLUE-4) compared to the best method using RGB video as input.


**Weaknesses:**

[1 – Writing quality]. In section 3.2, some analysis is confusing, and the conclusion seems to be the author's subjective thoughts. And in the title of table 2, ‘the inconsistent punctation bug’ is confusing.

[2 - method performance on CLS-Daily]. On the CLS-Daily dataset, MonoSLT performs poorly, lagging behind several sota models.

[3 - Model evaluation issues]. The paper also mentions that although it alleviates the problem of faithfulness in SLT, there are no suitable metrics to measure it. The author still uses BLEU and ROUGE for evaluation


**Questions:**

1. At the end of line 145, S2T model should be C2T model. This may be a spelling error.

2. In the title of table 2, the inconsistent punctation bug appeared, which confused me.  I don't know the meaning of this phrase.

3. On the CSL-Daily dataset, the performance of MonoSLT is somewhat lacking. Can you further improve the performance of MonoSLT on CLS-Daily?


**Limitations:**

1.Find or create proper metrics to quantitatively evaluate the faithfulness of SLT models.

2.You said that the CLS-Daily dataset provides more precise gloss annotations, which leads to other models with lower SLR performance being able to achieve better SLT results This also leads to your model not performing as well as some models on the CLS-Daily dataset. As the sign language dataset becomes larger and more accurate, your model may not be as good as other models. I think this is worth considering.

---

> ### Author Rebuttal · Authors · 2023-08-08
>
> **Q1. About the inconsistent punctation bug.**
>
> Thank you for bringing this up. During experiments, we find the tokenization process of the original implementation always predicts punctuation marks in English, which greatly affects the evaluation results, especially for longer n-gram. For example, ',' / '?' / '!' / ':' are punctuation marks in English, and '，' / '？' / '！' / '！' are punctuation marks in Chinese. We also re-evaluate the released models of MMTLB[12] **in the "global" response**. We will clarify relevant statements and update the results in Table 2.
>
> **Q2. About the poorly performance on CSL-Daily.**
>
> Thanks for your suggestion, it pushes us to figure out the performance gap between MMTLB[12] and the used baseline. As we mentioned in the main paper, this submission only uses the translation module of MMTLB, and we assume the used feature extractor (GCN+Conv1D+BiLSTM) is a universal structure for all forms of translation. To better leverage the precise gloss annotations provided by CSL-Daily, previous works (MMTLB and 2S-SLT[15]) take the gloss probabilities as the input to the VL-mapper and initialize the MLP layer with pretrained gloss embedding (presented in the supplementary of MMTLB). We follow this setting, but the BiLSTM layer will generate over-confident predictions due to its large temporal receptive field, which hinders the leverage of visual information and makes the model easily overfitting. Therefore, we only train V2T and G2T jointly on CSL-Daily and update relevant results as stated **in the "global" response**. With this modification, the gap between MMTLB and the used baseline is narrowed, and MonoSLT shows competitive performance with skeleton inputs only.
>
> **Q3. About the evaluation metrics for faithfulness.**
>
> Thanks for your constructive comments on the evaluation metrics. As we mentioned **in the “global” response**, we suggest evaluating the over-translation and under-translation with BLEU-1 and ROUGE-1, respectively. We also evaluate the proposed method on the subsets of different universal part-of-speech tags. Experiment results show that almost all tags benefit a lot from the proposed MonoSLT, especially for ADP, ADV, and PRON, which indicates the proposed method is effective in increasing faithfulness.
>
> **Q4. The generalization ability of the proposed method when datasets become larger and more accurate.**
>
> Thanks for your constructive feedback on the generalization on the larger scale dataset. Based on the manner of collecting data, recent sign language datasets can be summarized into two categories, some datasets (Phoenix14T[6], [R1], and [R2]) are collected from the broadcast or the Internet, and provide coarse-grained or automatically generated annotations, the others (CSL-Daily[11], R3) are collected by first designing the sign language corpus and then record sign videos from invited signers. The former way can easily collect large-scale datasets with imprecise annotations and is closer to real-world scenarios, while datasets collected in the latter way are under several constraints (e.g., the camera pose and background are kept the same in CSLDaily). From this perspective, the proposed method tries to leverage the alignment nature of sign video data rather than precise annotations, which indicates that it is more suitable for large-scale datasets. Moreover, it is worth noting that the computing cost also increases along with the size of the dataset, and the proposed skeleton-based method provides an effective way to quickly verify the effectiveness of SLU methods on large-scale datasets.
>
> [R1] Samuel Albanie, Gül Varol, Liliane Momeni, Triantafyllos Afouras, Joon Son Chung, Neil Fox, and Andrew Zisserman. Bsl-1k: Scaling up co-articulated sign language recognition using mouthing cues. In ECCV, 2020.
>
> [R2] Uthus D, Tanzer G, Georg M. YouTube-ASL: A Large-Scale, Open-Domain American Sign Language-English Parallel Corpus[J]. arXiv preprint arXiv:2306.15162, 2023.
>
> [R3] Amanda Duarte, Shruti Palaskar, Lucas Ventura, Deepti Ghadiyaram, Kenneth DeHaan, Florian Metze, Jordi Torres, and Xavier Giro-i Nieto. How2Sign: A large-scale multimodal dataset for continuous American Sign Language. In CVPR, 2021.
>
> **Q5. About the Writing quality.**
> Thanks for your constructive comments about the analysis in section 3.2. We agree with you that some conclusions are subjective and not well-supported, many possible factors lead to overfitting, and the lack of faithfulness is only one of them. Some conclusions here should be assumptions that motivate us to explore the leverage of visual signals in SLT. We have fixed typos and will carefully revise the relevant analysis to make it more thorough.

---

### Official Review · Reviewer_T3bw · 2023-07-07

**Soundness:** 3 good
**Presentation:** 3 good
**Contribution:** 2 fair
**Rating:** 5
**Confidence:** 5

**Summary:**

This work mainly studies the faithfulness issue in SLT (i.e., whether the SLT model captures correct visual signals). The study identifies imprecise glosses and limited corpora as the main factors contributing to limited faithfulness. In order to mitigate this issue, this work proposes a framework called MonoSLT, which leverages the shared monotonically aligned nature among SLT subtasks. This framework incorporates alignment and consistency constraints. Experiments demonstrates the effectiveness of the proposed method.

**Strengths:**

This paper is well-written and well-organized.

This work performs in-depth analysis on the previous works and the association among SLT-relevant tasks.

The overall performance is promising and shows notable performance gain over the baseline.

**Weaknesses:**

The main focus of this work is on the concept of faithfulness in spoken language translation (SLT). However, a notable limitation of the study is the absence of quantitative metrics to evaluate faithfulness. While the authors acknowledge this limitation in the paper's discussion of limitations, it remains a drawback. It would be beneficial for the authors to provide further clarification on this issue, perhaps by suggesting potential quantitative metrics that could be used to assess faithfulness in future research.

It is suggested that the proposed framework be compared with VAC, as they share similar components such as consistency loss and visual module constraints.

Regarding the discrepancy in length between the embeddings produced by the visual GCN module and the gloss module, it is essential to understand how the code-switching module handles this challenge. The authors should provide clarification or explanation on how the code-switching module addresses this issue.

**Questions:**

See the comments in Weakness.

**Limitations:**

It is better to design a suitable metric to evaluate faithfulness in SLT.

---

> ### Author Rebuttal · Authors · 2023-08-07
>
> **Q1: About the further clarification on the quantitative metrics.**
>
> Thanks for the insightful comment about the metric, and it pushes us to think about the evaluation metrics of faithfulness more deeply. As we mentioned **in the “global” response**, we suggest evaluating the over-translation and under-translation with BLEU-1 and ROUGE-1, respectively. Moreover, we believe the universal tokenization approach and the alignment between visual features (which may need extra annotations) and the prediction can further improve the interpretability of the SLT model.
>
> **Q2. About the comparison with VAC and the technical contributions.**
>
> Thanks for your suggestion. In fact, the design of preliminary experiments in Table 4 is inspired by VAC. Technically speaking, there is little difference in the constraint design between the proposed method and VAC: both adopt a visual module constraint and a consistency loss, and the main difference is that VAC needs pair-wise alignments, and we select two subtasks based on their characteristic and incorporate them as a single task to simplify the training process. Therefore, from the perspective of constraint design, the proposed method can be regarded as a simplified version of VAC, and we empirically infer the performance should be comparable. However, the proposed method is significantly different from VAC in the context of sign language understanding. We summarize the main differences and the technical contributions below.
>
> 1. **The faithful aspects of SLU models.** As stated in the "global" response, VAC can be regarded as a typical approach to improve the faithfulness of the visual encoder, and both constraints are designed for improving the visual encoding ability of the SLU models. However, this submission keeps the visual encoder the same and attempts to improve the faithfulness of the translation module, which tries to improve the translation process with the help of multiple subtasks. Besides, the proposed alignment constraint can also be applied to the "self-alignment" of visual features, which is verified in the experiments on CSL-Daily **in the "global" response**. Compared to the constraint designs, we believe the proposed framework is more important. We choose a VAC-style constraint design to validate the effectiveness of the proposed framework, and other alignment methods are also potentially feasible (e.g., contrastive-based methods).
>
> 2. **Explore the relationship between SLR and SLT**. Recent SLR and SLT studies are nearly independent, as we mentioned in the introduction, SLR approaches often validate their generalization ability on SLT tasks, and SLT approaches often adopt the pretrained SLR models and focus on the translation module designs. There exists a research gap between SLR and SLT works: whether the visual features are leveraged sufficiently. For example, as shown in Table 1 of the main paper, TwoStream-SLT [15] with the ensemble of multi-modality models (both skeleton sequence and video with three different random seeds) achieves much better SLR performance than with skeleton sequence only (-9.42% WER, from 27.14% to 17.72%), but it achieves comparable SLT performance (+0.97 BLEU-4, from 27.98% to 28.95%) under these two settings. It seems that the translation performance tends to be saturated, and this submission shows there is still a long way to make the SLT methods applicable.
>
> 3. **Explore the robustness of the SLT models**. The main difference between SLR and SLT is the different language habits, and this submission attempts to reveal the importance of faithfulness beyond performance: if the translation results change the meaning of the gloss sequence, the high BLEU-4 performance is meanless and the SLR model is a more practical choice than SLT. We hope this submission can inspire more works considering the robustness of SLT models.
>
> **Q3. How the code-switching module handles the length discrepancy.**
>
> Thanks for your suggestion, the visual sequence $V=(v_1, \cdots, v_T)$ and the gloss sequence $G=(g_1,\cdots,g_N)$ have different lengths, which prevent the alignment between the visual and semantic embeddings. We first estimate the alignment path $\hat{\pi}=\arg\max_\pi p(\pi|X, G)$ with the maximal probability [41], and then obtain the corresponding range $[s_i, e_i)$ for each gloss $i$, where $s_i$ and $e_i$ denote start and end frame indexes. Then we randomly sample gloss $i$ with the probability of $\beta$ and replace the corresponding visual embeddings $\mathcal{P}(v_i), i \in [s_i, e_i)$ with $\mathcal{E}(G_i)$ (i.e., replace all frames that are recognized as gloss $i$ with its semantic embedding). For gloss embedding that uses byte pair encoding, we average embedding of its subunits as the gloss embedding $\mathcal{E}(G_i)$. The matrix form is presented in Equ. 3. We will clarify the relevant statements and provide several examples in the supplementary.

---

### Author Rebuttal · Authors · 2023-08-07

**Thanks for the constructive suggestions and insightful comments from all reviewers, they push us to think about the faithfulness in SLU more deeply. Our review responses for common questions are summarized below.**

**Q1. Potential quantitative metrics for faithfulness**

Inspired by previous works in NMT [R1], we summarize unfaithfulness in SLU to three problems: unfaithful visual encoder, over-translation, and under-translation. This submission keeps the visual encoder the same and does not dig into the faithfulness of the visual encoder. We provide potential quantitative metrics for the latter problems as below.

**Over-translation: some words are unnecessarily translated for fluency.** The over-translation problem happens when the SLU model takes the wrong visual information or relies too much on the target-side context and the implicit language model. However, it is hard to evaluate the utilization of visual information, and the lack of ground-truth alignment also makes it difficult to evaluate the inference process of the translation module. Therefore, we attempt to evaluate the over-translation problem from the predictions.

Compared to proposing a new metric, we suggest using existing metrics (specifically, the BLEU-1) to evaluate the over-translation problem. As mentioned in the original paper [R2]:
> A translation using the same words (1-gram) as in the references tends to satisfy adequacy. The longer n-gram matches account for fluency.

Different from longer n-gram metrics, BLEU-1 evaluates the precision of word-wise prediction without regard to fluency. To quantitatively evaluate the faithfulness of the proposed method, we further calculate the BLEU-1 with different universal part-of-speech tags:
$$
\text{BLEU-1(tag)}=\frac{\sum_{C\in{HYP}}\sum_{word\in C, \tau(word)==tag} Count_{clip}(word) }{\sum_{C\in{HYP}}\sum_{word\in C, \tau(word)==tag} Count(word) }，
$$
where $\tau(\cdot)$ is a model to obtain the part-of-speech tag of a word.

**Under-translation: some visual information is ignored or mistakenly understood.** The under-translation problem happens when the SLU model ignores the critical visual information or takes wrong visual information for translation. Similar to the over-translation problem, we suggest using ROUGE-1 to evaluate the under-translation problem. The ROUGE-1 scores with different universal part-of-speech tags are calculated by:

$$
\text{ROUGE-1(tag)}=\frac{\sum_{C\in{REF}}\sum_{word\in C, \tau(word)==tag} Count_{clip}(word) }{\sum_{C\in{REF}}\sum_{word\in C, \tau(word)==tag} Count(word) }，
$$

The evaluation results with BLEU-1 and ROUGE-1 on the Dev set of Phoenix14T are presented in the following table. The completed table and results on the Test set can be found in the attached PDF.

||BLEU-1|||ROUGE-1|||
|-|:-:|:-:|:-:|:-:|:-:|:-:|
||Baseline|MonoSLT|#Word|Baseline|MonoSLT|#Word|
|ADP|0.561|**0.612**|997/1000|0.545|**0.593**|1037|
|ADV|0.455|**0.494**|1333/1274|0.419|**0.431**|1479|
|DET|0.563|**0.568**|554/572|0.514|**0.528**|615|
|NOUN|0.617|**0.643**|1480/1469|0.591|**0.610**|1535|
|PRON|0.484|**0.493**|428/434|0.516|**0.547**|397|
|PROPN|**0.444**|0.239|160/381|0.445|**0.555**|164|
|VERB|0.326|**0.341**|469/472|0.310|**0.335**|484|
|Overall|0.563|**0.570**|6989/7205|0.536|**0.559**|7339|

As shown in the tables, almost all tags benefit a lot from the proposed MonoSLT, especially for ADP, ADV, and PRON, which indicates the proposed method is effective in increasing faithfulness. This result is roughly coincident with the previous finding in NMT [R3] that the increase in faithfulness for function words is much more than that of content words. The linguistic structure of sign language is complicated yet fascinating, we hope the proposed method and metrics can inspire further works.

R1. Modeling coverage for neural machine translation. ACL, 2016.
R2. Bleu: a method for automatic evaluation of machine translation. ACL, 2002.
R3. Measuring and improving faithfulness of attention in neural machine translation. EACL, 2021.

**Q2. Punctuation bug and poor performance on CSL-Daily**

To provide a more thoughtful evaluation on the bug-free dataset, we re-evaluate the released models of MMTLB[12] and 2S-SLT[15] by adding an extra line in the evaluation script provided by [12]:
```
txt_hyp = txt_hyp.replace(',', '，').replace('?', '？').replace('!', '！').replace(':', '：')
```
where ',' / '?' / '!' / ':' are punctuation marks in English, and '，' / '？' / '！' / '！' are punctuation marks in Chinese. The tokenization process of the original implementation always predicts punctuation marks in English, which greatly affects the evaluation results, especially for longer n-gram. The re-evaluate results are presented in the following table.

||Dev Set|||Test Set||||||
|:-:|:-:|:-:|:-:|:-:|:-:|:-:|:-:|:-:|:-:|
||R|B4|WER|R|B1|B2|B3|B4|WER|
|MMTLB[12]|55.76|27.43|30.6|56.06|56.61|43.66|34.31|27.51|30.3|
|2S-SLT[15]|58.24|29.18|25.4|**58.62**|**58.64**|**45.77**|**36.39**|**29.55**|**25.3**|
|Baseline|54.86|27.00|30.1|55.37|56.03|42.82|33.62|27.02|*29.1*|
|MonoSLT|55.79|28.09|29.9|*56.25*|*57.28*|*44.15*|*34.89*|*28.19*|29.2|

As suggested by reviewers, we attempt to figure out why MonoSLT performs poorly on CSL-Daily. We find that adopting visual logits as input can achieve better performance than contextual logits (27.0\% v.s. 25.9\%). One possible reason is that we initialize the MLP layer with pretrained gloss embedding as MMTLB[12] does, and the overconfident predictions of BiLSTM layer hinder the leverage of visual information. Therefore, we only train V2T and G2T jointly on CSL-Daily, and the final supervision is formulated as：

$$
L = L_{CTC}^{V}+\lambda_C L_{SLT}^V + \lambda_{CS} L_{SLT}^{CS} + \lambda_c(D_{KL}(p_V||p_{CS})+D_{KL}(p_{CS}||p_{V})).
$$

The corresponding results are updated in the table above. With this modification, MonoSLT achieves better performance than MMTLB with comparable WER, which indicates the effectiveness of MonoSLT.

---

### Decision · Program_Chairs · 2023-09-21

**Decision:**

Reject

**Comment:**

The original submission of this paper did not clearly define the measures associated with faithfulness. We thank the authors to submit a rebuttal with new proposed metrics for faithfulness. Having them in the original submission would have helped reviewers. The technical novelty of the approach seems to be about integrating subtasks into one framework. Since the NLA-SLR paper was published officially after the submission of this paper, I downplayed its impact, but the idea of mixup is not novel in itself, while it may have some novelty for sign language translation. Since the proposed framework is in big part based on VAC, it also reduces its technical novelty. The main novelty seems to be in the analyses and exploration. After reading all discussions, I am leaning towards reject.